# REVOLUTIONIZING AI COMPANION IN FPS GAMES

## ABSTRACT

Traditionally, players in first-person shooter (FPS) games have been limited to communicating with AI companions using simple commands like "attack," "defend," or "retreat" due to the constraints of existing input methods such as hotkeys and command wheels. One major limitation of these simple commands is the lack of target specificity, as the numerous targets in a 3D virtual environment are difficult to specify using existing input methods. This limitation hinders players' ability to issue complex tactical instructions such as "clear the second floor," "take cover behind that tree," or "retreat to the river." To overcome this limitation, this paper introduces the **AI Companion with Voice Interaction (ACVI)**, the first-ever AI system that allows players to interact with FPS AI companions through natural language. Deployed in the popular FPS game *Arena Breakout: Infinite*, this revolutionary feature creates the most immersive experience for players, enabling them to work with human-like AI. ACVI is not confined to executing limited commands through simple rule-based systems. Instead, it allows players to engage in real-time voice interactions with AI teammates. By integrating various natural language processing techniques within a confidence-based selection framework, it achieves rapid and accurate decomposition of complex commands and intent reasoning. Moreover, ACVI employs a multi-modal dynamic entity retrieval method for environmental perception, aligning human intentions with decision-making elements. It can accurately comprehend complex voice commands and delivers real-time behavioral responses and vocal feedback to provide close tactical collaboration to players. Additionally, it can identify more than 17,000 objects in the game, including buildings, vehicles, grasslands, and collectible items, and has the ability to accurately distinguish different colors and materials.

## 1 INTRODUCTION

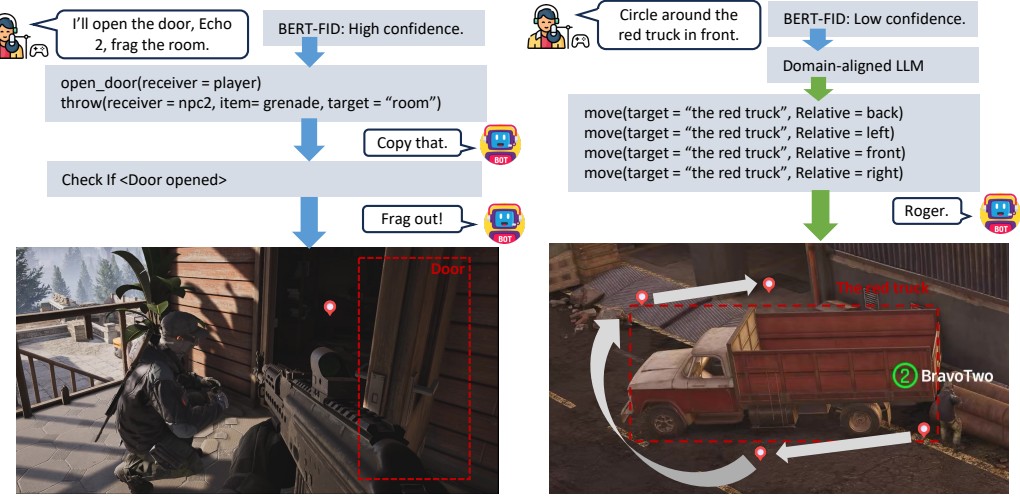

Figure 1: ACVI is the first-ever open-interaction and real-time voice-operated AI companion system for commercial 3D FPS games.

Over the past few decades, video games have attracted billions of players, and the role of game AI has become increasingly important (Ontanón et al., 2013). Game AI in board games like Go (Silver et al., 2016) and real-time strategy games like StarCraft II (Vinyals et al., 2019) have already reached expert player levels. However, most game AI systems focus on competing with players, and research on AI companion systems in games has been largely overlooked (Gao et al., 2023; Ashktorab et al., 2020). In complex Multi-player Online Battle Arena (MOBA) (Berner et al., 2019; do Nascimento Silva & Chaimowicz, 2017) or First-person Shooter (FPS) games (Lample & Chaplot, 2017), players need the companionship of teammates, and meticulous cooperation between players and teammates is crucial for achieving goals and an immersive experience. Therefore, attaining reliable human-AI collaboration in games and creating an AI companionship system that can understand human instructions and provide quick and accurate assistance is a promising research field.

In traditional FPS games (Tong et al., 2011), players typically use simple commands such as hotkeys and command wheels to interact with AI companions, for example, by sending quick commands to attack, defend, or retreat. The primary limitation of these simple commands is the lack of target specificity, as the numerous targets in a 3D virtual environment are difficult to specify using existing input methods. This limitation hinders the player's ability to issue complex tactical commands such as "clear the second floor", "take cover behind that tree", or "retreat to the river", creating a gap in human-AI collaboration and a poor gaming experience.

To address this problem, we propose AI Companion with Voice Interaction (ACVI), the first-ever FPS AI companion that can interact with players through natural language, provide tactical collaboration, understand the game environment, and identify in-game objects with precision. Deployed in the popular FPS game *Arena Breakout: Infinite* (Tencent, 2024), this revolutionary feature aims to create the most immersive human-AI collaboration experience for its players. ACVI is not limited to executing a finite set of commands through a simple rule system but allows players to interact with the AI teammate in real-time voice interactions, executing a large number of open-ended commands. The ACVI system follows a three-part pipeline: instruction reasoning, environmental perception, and decision execution. First, we proposed **BERT-FID** (BERT designed for Fuzzy Intent Detection) to handle the decomposition of complex player instructions and intent reasoning. It offers a confidence-based selection mechanism that integrates the capabilities of a smaller model (BERT) and a more powerful large language model (LLM) for switching inference. This enables the system to respond quickly to simple commands while distributing fuzzy commands to the LLM for further understanding and analysis, achieving accurate comprehension. This approach effectively reduces the system's deployment costs and inference latency while maintaining comprehension accuracy. Then, the player's commands are converted into action and target description templates. We introduce a novel multi-modal dynamic entity retrieval framework for environment perception, matching the player's commands with real-time game observations and assets. Ultimately, the player's complex commands are broken down into multiple atomic instructions, and integrated with various environmental asset information that needs to be considered. The AI companion can instantly execute strategies through behavior trees and provide feedback to the player. Fig. 1 demonstrates the capability of ACVI in completing tasks. A full match visualization video is released on our homepage.

We summarize the contributions of our work as follows:

- We introduce BERT-FID, a confidence-based selection mechanism specifically designed for fuzzy intention detection which integrates the advantages of both small models (BERT) and more powerful models (LLM), achieving competitive accuracy about 88.6% close to solely LLMs with a 4-fold increase in inference concurrency.

- We present a multi-modal dynamic entity retrieval method for environmental perception and game asset recognition, which can accurately retrieve over 17,000 objects with different colors and materials in the game scene, enabling fine-grained control over the companions. Our enhancements method can identify 96.6% dynamic entities in the game with a 5-fold increase in inference concurrency.

- By integrating instruction reasoning, environmental perception, and decision execution pipelines, we develop ACVI system, the first voice-interactive FPS game AI companion in the *Arena Breakout: Infinite*, revolutionizing AI companions in commercial FPS games. The flexible design of ACVI enables a low-cost, real-time inference execution process, enables open-ended voice interactions, capable of comprehending users' open-ended commands. The flexible design of ACVI

enables a low-cost, real-time inference execution process, get the action response from text request approximately 613 ms, resulting in a 77% cost reduction compared to baselines.

## 2 BACKGROUND AND RELATED WORK

*Arena Breakout: Infinite* is a tactical extraction FPS game developed and globally published by Tencent Games. Its core gameplay offers players the freedom to choose their own playstyle. Aggressive players with high-quality equipment may actively engage in combat and loot the bodies of fallen enemies for weapons and armor. In contrast, passive players can focus on looting the map and avoiding crossfire to complete in-game quests or gather more collectibles before extraction. However, the struggle to reach a consensus on whether to engage in combat can make it challenging to achieve a positive cooperative experience with randomly matched human teammates. This scenario presents opportunities for teaming up with AI companions.

The integration of AI companions in gaming has recently attracted significant attention (Gallotta et al., 2024; Xu et al., 2024; Park et al., 2023; Ma et al., 2023), particularly concerning roles involving non-player characters (NPCs) and player assistance. Shao et al. (2023) introduced Character-LLM, a framework designed to train LLMs to simulate specific characters, thereby enhancing role-playing experiences. Zhang et al. (2024) developed a text-to-game engine that utilizes LLMs to transform text inputs into dynamic RPGs, automatically generating game content such as storylines and mechanics in real time. Rao et al. (2024) explored the application of LLMs in powering NPCs in 3D games like Minecraft, where players collaborate with LLM-driven agents to complete tasks. While existing studies primarily focus on conversational interactions and the generation of narrative text (Gursesli et al., 2023), they often overlook the enhancement of game-state and visual understanding necessary for improving in-game collaboration. In commercial games, NetEast & Entertainment (2020); Interactive (2023) are also working on developing more complex AI companion systems, but they are struggle to decomposing a wider range of player instructions and a precise ability to perceive game scenarios. AI companions frequently struggle with spatial reasoning (Team et al., 2022) and planning—crucial elements (Wang et al., 2024) in many games, especially FPS games, which depend on precise spatial reasoning and strategic planning (Ontanón et al., 2024).

Intent recognition and slot filling are critical tasks for AI companion system. Early methods (Huang et al., 2015; Krizhevsky et al., 2017; Liu & Lane, 2016) relied on traditional machine learning, which limited in handling complex linguistic phenomena. With the advent of pre-trained language models, BERT-based methods (Vaswani et al., 2017; Devlin, 2018; Chen et al., 2019; Comi et al., 2023) has shown outstanding performance in handling downstream tasks. Recently, LLMs have achieved remarkable generalization capabilities. Brown et al. (2020) and Patel et al. (2023) possess powerful generative and comprehension abilities, making zero-shot learning feasible, greatly reducing the dependency on labeled data. The work by (Zhang et al., 2021b; 2022; 2021a) discusses the classification of open intents in real dialogue scenarios, aiming to discover the decision boundaries required in real-world applications. Some recent works (Leviathan et al., 2023; Lin et al., 2023; Chen et al., 2023; 2024) explore a collaborative paradigm between small and large models to accelerate inference and reduce deployment costs across various text-related tasks. Inspired by these works, we have equipped BERT with the capability to perform fuzzy intent detection. By leveraging the stronger generalization capabilities of large language models (LLMs), we can infer these new intents more effectively.

Image-text retrieval (ITR) is a crucial cross-modal task aimed at bridging visual and textual information. Encoding image and text features independently is a straightforward approach. Frome et al. (2013) and Faghri et al. (2018) employ classical vision and text encoder models and subsequently align them through similarity calculations.The emergence of large-scale cross-modal pre-training techniques leverage large-scale web data for the feature extractors, demonstrating impressive performance on ITR tasks. CLIP (Radford et al., 2021; Yang et al., 2022a) by OpenAI leverages large-scale image-text pairs for contrastive learning, greatly enhancing image-text retrieval performance. Dual-stream models are well-suited for this application scenario because they allow for the offline storage of decoded images, resulting in higher computational efficiency during real-time inference.

Figure 2: **Overview of ACVI.** Players can cooperate with the ACVI agent in combat and freely speak into the microphone and get behavior response with voice feedback.

# 3 OVERVIEW OF ACVI

As shown in Fig. 2, players can cooperate with the ACVI agent in combat and freely speak into the microphone. After processing by the general Automatic Speech Recognition (ASR) (Malik et al., 2021; Wang et al., 2017) module, the spoken content is converted into textual information. Players can cooperate with the ACVI agent in combat and freely speak into the microphone. We implement a confidence-based method for the instruction reasoning, which can hand over some intricate instructions that the BERT-FID classifier cannot handle to the domain-aligned LLM. After parsing the player's action and target description, the multi-modal dynamic entity retrieval module is activated to obtain specific 3D coordinate information within the game engine. Once the parsed command sequence is sent to the game, the AI companion can immediately begin executing actions based on behavior trees and generate voice feedback through the real-time text-to-speech (TTS) (Kumar et al., 2023; Kim et al., 2021). These modules are interconnected to form the complete ACVI system, enhancing interpretability and facilitating individual optimization, analysis, and upgrades of each component. As detailed in the subsequent sections, the enhancements in instruction reasoning and scene recognition are core technologies that contribute to the success of ACVI.

# 4 CONFIDENCE BASED INSTRUCTION REASONING

We introduce a fast-processing component called BERT-FID to filter out clear tactical instructions. Then, then integrate BERT-FID with a domain-aligned LLM, enabling our system to effectively manage complex tactical instructions that involve contextual reasoning, task decomposition, reference resolution, and more. This approach allows ACVI to swiftly execute simple tasks while also addressing more intricate instructions.

## 4.1 BERT WITH FUZZY INTENT DETECTION

| Command | Hold | position | at | the | red | door | I | will | grab | the | loot |
|---|---|---|---|---|---|---|---|---|---|---|---|
| Segment | B-seg | I–seg | I–seg | I–seg | I–seg | I-seg | B–seg | I-seg | I-seg | I-seg | I-seg |
| Entities | O | O | O | O | B-entity | I-entity | B–name | O | O | O | B-entity |
| Syntactic | B-subject | O | O | O | B-object | I-object | B–subject | O | O | O | B-object |
| Intent | Tactic-move | | | | | | Tactic-move | | | | |
| Receiver Type | Null | | | | | | Player | | | | |

Figure 3: **Multi-task Label Structure.** The label structure designed for a multi-task training method involving command segmentation, intent classification, and named entity recognition.

**Multi-task Label Structure.** In the context of FPS games, player commands may need to be divided into a sequence of executable actions for the behavior tree, and it is essential to extract the entity targets for each sub-command's intent. The structure of data labels is shown in Fig. 3. The segment labels in the second row are used for handling common segmentation tasks, where "B-seg" denotes the beginning of a sub-command. The labels for the named entity recognition task identify the entity types and syntactic tagging. The intent classification labels encompass both the types of intent and the classification of their subjects. Specifically, where a player issues the command, "Hold position

at the red door, I will grab the loot", the BERT model can understand and separate this input and output the intent as "move", the target location as "the red door".

**Model Architecture.** As shown in Fig. 4, we rely on Bidirectional Encoder Representations from Transformers (BERT) (Vaswani et al., 2017; Devlin, 2018), which is based on the self-attention mechanism, as the pre-training representation model in the feature extraction layer. In downstream tasks, we introduce command segmentation to decide whether each token starts a sentence, thereby obtaining executable sub-commands. We integrated the token-level features of the divided sub-commands through an attention mechanism to form new features. These newly formed features serve as the input for subsequent hierarchical intent classification and named entity recognition.

The named entity recognition can be regarded as a sequence classification task, which can utilize the Conditional Random Field (CRF) (Lafferty et al., 2001) to consider the context information of the input sequence to accurately calculate the conditional probability distribution. In the hierarchical intent classification, and a regressive embedding mechanism is introduced between the levels to complete a more comprehensive feature extraction.

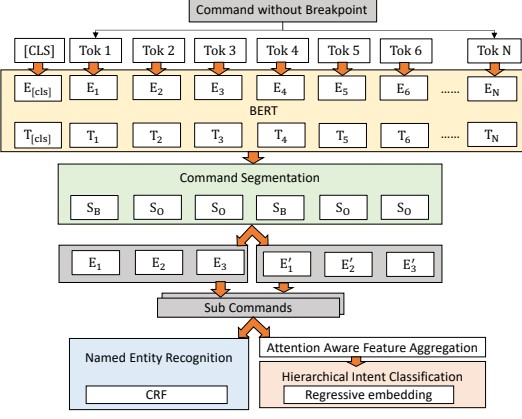

Figure 4: Model Architecture.

We utilize this BERT base model architecture to perform joint training on three tasks, effectively standardizing high-dimensional categories under the same data distribution. These tasks include a text segmentation task with a maximum length of 128 tokens, named entity recognition (NER) for six different target entities, and a hierarchical intent classification task with a three-layer intent recognition network encompassing a total of 25 categories.

**Considering Fuzzy Intent Detection in Training.** In order to allow the model to fully leverage the predicted confidence, we introduce entropy optimization objective for unsupported intent classification head in hierarchical classifier. This method can effectively enhance the model's capability to detect the fuzzy intent while also alleviating the strain of training.

Given a batch of $N = N_C + N_e$ samples, $N_c$ supported data use cross-entropy to calculate the general loss function, while the remaining $N_e$ unsupported data utilize maximization of entropy to increase the uncertainty of the probability vector. The loss function of this method is as follows:

$$\mathcal{L} = -\frac{1}{N_c} \sum_{i=1}^{N_c} \sum_{j=1}^{C} w_j y_{ij} \log p_{ij} - \lambda \frac{1}{N_e} \sum_{i=1}^{N_e} \left( \log C - \sum_{j=1}^{C} p_{ij} \log p_{ij} \right) \tag{1}$$

where $C$ is the number of categories for one head; $y_{ij}$ is the one hot encoded label for the $j$-th category in the $i$-th sample; $p_{ij}$ is the probability vector indicating the likelihood of the $i$-th sample belonging to the $j$-th category; $w_j$ is the weight of the $j$-th category, which is determined based on the statistical results of intent labels $\frac{N_c}{n_i}$, where $n_i$ is the number of $j$-th category samples. This method increases the weight of sparse labels and reduces the weight of dense labels, especially needed in the unbalanced intent distribution in our scene.

This approach enables the model to reduce the confidence level for any category when encountering data beyond the predicted categories. Especially in the hierarchical intent classification we have set up, the true labels at the first level determine which prediction heads need to be activated at the second level. For instance, only when the first intent label is "throw", do we need to predict the next level of item head as grenade, smoke, food, or water. If this prediction head is not activated, we will minimize the confidence in any specific category by maximizing entropy regularization. The pseudocode can be found in Appendix A.1.

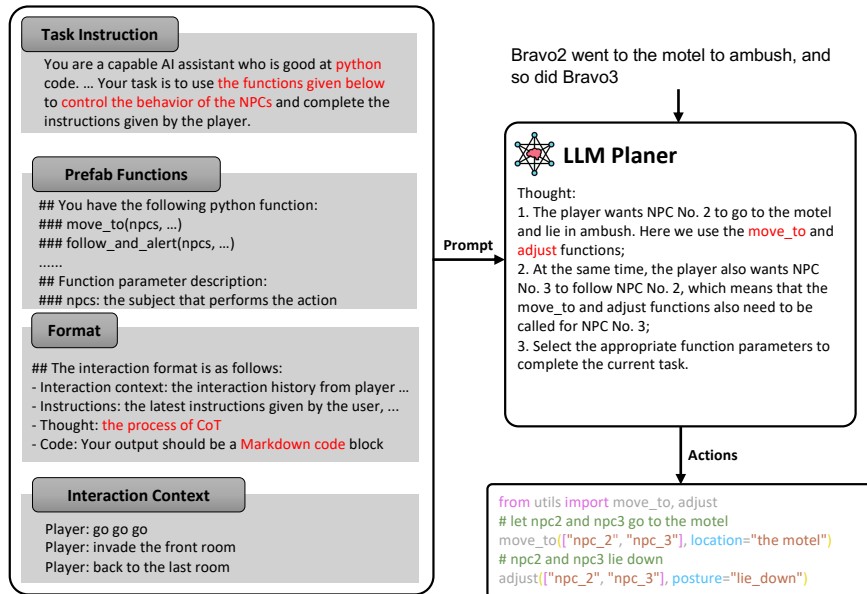

Figure 5: **LLM for Actions Planning.** The LLM Planner synthesizes Python code to call these functions to orchestrate the actions of ACVI agents, reasoning and acting to address complex tactical tasks.

## 4.2 DOMAIN-ALIGNED LLM

To enable the ACVI agents to accurately execute complex and ambiguous commands that BERT-FID achieve low confidence, we have fine-tuned a lightweight LLM using our in-house SFT dataset to better align with the preferences of real-life tactical companions.

As shown in Fig. 5, we adopt a code generation approach with LLMs to control the actions of ACVI agents. We find that Python is more suitable for manipulating the ACVI agents due to its alignment with the training corpora of most existing LLMs. Moreover, compared to having LLMs output in JSON format, Python code structures offer more flexible expressive capabilities, enabling the representation of common code constructs such as conditional and loop structures, which are not inherently supported by the JSON format. We designed a series of atomic tasks (implemented by behavior trees, e.g. move, attack, open_door), which are represented as Python functions in LLM. The LLM Planner is trained to generate Python code that invokes these functions, enabling the agents to reason and act in order to address complex tactical tasks.

We obtain the expected actions of FPS game agents in response to various player commands using a state-of-the-art general-purpose LLMs. These actions are then further validated by human annotators. The command-actions pairwise dataset constructed in this manner is used for the supervised fine-tuning of the lightweight LLM.

## 5 MULTI-MODAL DYNAMIC ENTITY RETRIEVAL

This section will elaborate on how to implement a real-time multi-modal scene recognition system capable of fine-grained understanding a large number of complex and diverse objects based on the player's field of view.

As shown in Fig. 6, we structure the 3D game assets dataset with tactical information such as coordinate positions, orientations, bounding boxes, and cover-points. Coordinate positions are used for ACVI to perform navigation. Orientation infor-

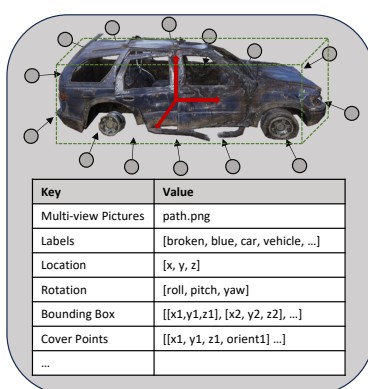

Figure 6: Game assets Structure.

mation help ACVI in understanding the physical spatial rela-
tionships of game assets. Bounding boxes are utilized for ACVI to perceive the size and range of physical assets. Cover points, obtained from the game's cover system, guide ACVI to find more reasonable positions in FPS game.

For the input entity description, we first conduct a similarity search against the embeddings of all entity labels in the scene. In cases where the number of retrieved entities is insufficient, we utilize a fine-tuned CLIP (Radford et al., 2021) model to perform a similarity search on the image embeddings of all entities in the scene, ultimately generating a list of candidate entities.

We take real-time dynamic game data into account for the fine ranking of candidate entities, such as player's position and orientation. As illustrated in Fig. 7, although entity3 shares the highest similarity with the human command, it is too far away. On the other hand, entity5 is the closest but has exceeded the range specified by the directional term. Therefore, the final search result is entity1, which comprehensively ranks highest when considering all factors.

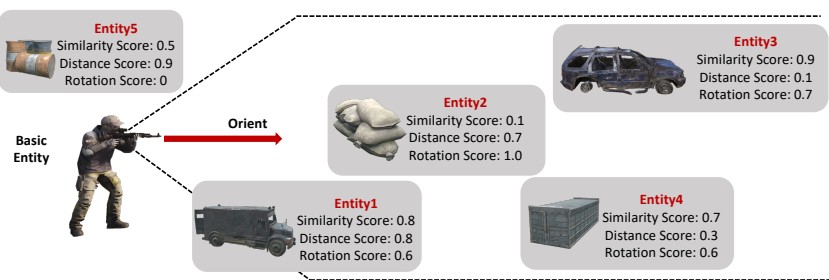

Figure 7: **Dynamic Ranking in Game.** We integrated the results of image-text retrieval with dynamic game information to conduct entity searches, identifying the object with the highest overall ranking as the player's intended target location.

Additionally, our dynamic entity search method can handle complex hierarchical relationships. For instance, if a player inputs: "Go behind the red sofa on the first floor of the motel and find a cardboard box" after command segmentation and named entity recognition in Section 4.1, we can establish a hierarchical search relationship. First, we locate the building "the first floor of the motel" then use that coordinate location as a basic point to find the secondary target "red sofa" and finally, using the sofa as a basic point, we search for the nearby "cardboard box."

Furthermore, since we have cover points data, we can also use the entity search method to instruct ACVI to hide at specific locations. Each object has multiple cover points, and each cover point has its corresponding orientation—for example, only when facing the entity can it act as a cover. In this search task, we need to prioritize the orientation relationship. By passing different weighted parameters for specific tasks, our entity search system can accomplish a rich variety of functions.

## 6 EXPERIMENTS

### 6.1 SETTINGS

**Evaluation Dataset**. To continuously assess the capabilities of ACVI and gather player data, we have established a human testing and evaluation system that leverages real-time feedback from players (ref to Appendix A.2). During gameplay, players can provide authentic feedback on their interactions with AI teammates using thumbs-up and thumbs-down buttons. The data collected from the thumbs-up button, which reflects the model's predicted outcomes, can be used directly as part of the evaluation dataset. The data from the thumbs-down button were annotated and incorporated into the evaluation dataset. In each game session, we can collect between 10 and 30 pieces of authentic data, and after 100 sessions with human players, we have gathered a total of 2,550 data points for our evaluation dataset.

**Evaluation Models.** ACVI has been developed in both Chinese and English. For different languages, only the base model, dataset, and LLM prompt need to be adjusted. Since the participants in

our user study were Chinese players, we will introduce the corresponding model settings for the experiment here. The BERT-FID model utilizes the "Chinese-lert-base" pre-trained model (Cui et al., 2022) and incorporates modules such as attention pooling and CRF for downstream tasks. Additionally, we perform cross-level feature fusion for hierarchical intent classification. The model was trained on 150,000 training samples for 20 epochs, requiring 3 hours on an A100 machine.

We fine-tuned a lightweight Qwen2-1.5B model (qwe, 2024) using full-parameter fine-tuning in our in-house SFT dataset. We initially collected 2000 player commands from our testing and annotation system. We employed an advanced generic state-of-the-art LLM to generate ACVI agents actions for these 2000 commands. Subsequently, human intervention was applied to meticulously refine the generated results to align with human preferences for AI companions behaviors. This curated set of 2000 high-quality data constitutes our supervised fine-tuning (SFT) dataset.

The environment recognition module of ACVI utilized "tencent-ailab-embedding" (Song et al., 2018) to calculate the text similarity, which contains 102M parameters. In order to save the runtime text embedding time, we saved 150,000 embedding vectors of approximate words from the 1156 game item tags, as a static word2vec model. The clip model is based on "Chinese-CLIP-ViT-Base-Patch16" (Yang et al., 2022b), which contains 86M visual parameters and 102M text parameters. We fine-tune the model on 10240 image-text data of 1277 game items for 10 epochs.

**Evaluation Metrics**. We first conducted tests on various solutions for the instruction reasoning module and the scene recognition module, ultimately performing comparative tests on the integration of both modules. For the instruction reasoning module, in addition to accuracy, we also report the "Final Decision Sample Count," which indicates the number of samples for which each model made the final decision during evaluation. This metric is crucial to assess which model contributes to the final inference results. To provide a comparative evaluation of both accuracy and efficiency, we present the concurrency of various methods by reporting the queries per second (QPS) metric from stress testing, along with the corresponding average and standard deviation of latency. This inference performance module was implemented on two NVIDIA GeForce RTX 3080 GPUs, while scene recognition was carried out using two NVIDIA Tesla T4 GPUs.

## 6.2 RESULTS

Table 1: The comparison of various methods in about instruction reasoning demonstrates that our approach, "BERT-FID with fine-tuned LLM", can route more samples to the LLM and effectively achieve higher accuracy than the pure BERT-based method.

| Method | Accuracy | Parameters | Final Decision Sample Count | |
| --- | --- | --- | --- | --- |
| | | | BERT | LLM |
| BERT | 0.714 | **104M** | 2550 | 0 |
| BERT-FID | 0.749 | | | |
| Qwen2-7B | 0.807 | 7B | 0 | 2550 |
| Qwen2-1.5B | 0.642 | 1.5B | | |
| Domain-aligned LLM | **0.908** | | | |
| BERT with Domain-aligned LLM | 0.831 | 1.6B | 2321 | 229 |
| **BERT-FID with Domain-aligned LLM** | 0.886 | | **2148** | **402** |

Note: the "Final Decision Sample Count" indicates the number of samples for which each model made the final decision during evaluation.

The results in Table 1 indicate that the accuracy is the lowest due to the presence of a significant amount of unsupported data in the evaluation dataset that the BERT classifier cannot recognize. The domain-aligned LLM method addresses these complex tasks more effectively than the general qwen2 model (qwe, 2024) which has not been aligned in our FPS AI companion scenario. The metric of final decision sample count indicates that the baseline method, "BERT with Domain-aligned LLM," made more erroneous high-confidence decisions, resulting in lower precision compared to our method, "BERT-FID with Domain-aligned LLM." Our approach was able to forward 15.8% of the data to the LLM for further evaluation, significantly enhancing accuracy.

The results in Table 2 demonstrates that the general CLIP model struggles to achieve satisfactory results in retrieving 3D game assets. However, after fine-tuning with in house asset labels, we were able to significantly improve the accuracy of text-image matching. We also present the impact of the

Table 2: Results on various retrieval range and various methods in environment recognition. By combining text matching with the Fine-tuned CLIP model, we can improve the accuracy across several retrieval range.

| Retrieval Range | Method | Pre. | Rec. | F1 |
|---|---|---|---|---|
| Top 5 | General CLIP | 0.459 | 0.4 | 0.429 |
| | Fine-tuned CLIP | 0.817 | 0.763 | 0.79 |
| | Text similarity | 0.904 | 0.873 | 0.888 |
| | **Text similarity with fine-tuned CLIP** | **0.926** | **0.89** | **0.908** |
| Top 10 | General CLIP | 0.578 | 0.521 | 0.549 |
| | Fine-tuned CLIP | 0.578 | 0.521 | 0.549 |
| | Text similarity | 0.93 | 0.913 | 0.921 |
| | **Text similarity with fine-tuned CLIP** | **0.953** | **0.932** | **0.943** |
| **Top 20** | General CLIP | 0.688 | 0.636 | 0.662 |
| | Fine-tuned CLIP | 0.923 | 0.904 | 0.913 |
| | Text similarity | 0.946 | 0.936 | 0.941 |
| | **Text similarity with fine-tuned CLIP** | **0.966** | **0.954** | **0.96** |

retrieval range on multi-modal retrieval. As the retrieval range increases, accuracy improves, and our method consistently achieves better accuracy. The scene perception module in ACVI combines text models with the Fine-tuned CLIP model, effectively enhancing accuracy about 96.6% in the range of top 20.

Table 3: Results from performance testing indicate that our method achieves approximately a four-fold increase in inference concurrency compared to LLM-based methods, and a five-fold increase compared to CLIP-based methods.

| Module | Method | QPS | Latency(ms) | |
|---|---|---|---|---|
| | | | Average | Standard Deviation |
| Instruction Reasoning | BERT-based method | 1672 | 412 | 33 |
| | Qwen2-7B | 46 | 423 | 212 |
| | Domain-aligned LLM | 245 | 392 | 121 |
| | BERT-FID with Domain-aligned LLM | 813 | 470 | 423 |
| Dynamic Entity Retrieval | CLIP-based method | 194 | 355 | 15 |
| | Text similarity | 2128 | 364 | 16 |
| | Text similarity with Fine-tuned CLIP | 1057 | 363 | 27 |

In the performance testing shown as Table 3, we conducted stress tests across various methods using the evaluation dataset on a consistent platform. LLM-based instruction reasoning methods generate action sequences of variable lengths, leading to higher variance compared to BERT-based classification models. Simple samples are effectively resolved with high confidence by smaller models, while complex samples require inference from larger models, resulting in greater variance in the collaborative reasoning model. Our method of instruction reasoning significantly enhances the system's concurrency, achieving a QPS within 500ms that is approximately 4-fold that of a pure LLM system. Our method of dynamic entity retrieval can achieve 5-fold concurrency compared to the pure CLIP system while ensuring that the vast majority of requests are responded 400ms. Improvements in instruction reasoning and scene recognition reduced inference deployment costs by approximately 75%.

Table 4: The overall evaluation of intent and environment recognition indicates that by integrating four model inferences, ACVI achieves an accuracy of 87.2% and a QPS of 916, with a average response time of approximately 613 ms, providing a high-precision, cost-effective solution for commercial games.

| Method | Accuracy | QPS | Latency(ms) | |
|---|---|---|---|---|
| | | | Average | Standard Deviation |
| BERT-FID and Text Similarity | 73.9% | **2050** | **599** | **79** |
| Domain-aligned LLM and Fine-tuned CLIP | **89.2%** | 210 | 614 | 150 |
| **ACVI's intent and environment recognition** | 87.2 % | 916 | 613 | 710 |

In the overall performance evaluation presented in Table 4, we compared the accuracy and inference performance of three different solutions. The intent and environment recognition method of ACVI achieved an accuracy of 87.2% and QPS of 916, with a response time of approximately 613 ms. Our approach efficiently processes simple samples quickly, whereas complex samples are handled with greater precision using larger models, resulting in a higher variance in inference. The intent and environment recognition method of ACVI achieved greater precision compared to the BERT text similarity solution and offered a more cost-effective alternative than the LLM with CLIP approach, reducing inference deployment costs by approximately 77%.

# 7    CONCLUSION AND FUTURE WORK

This paper introduces ACVI, a low-cost, high-precision solution for AI companions with advanced natural language understanding and visual perception capabilities. It serves as a revolutionary example of how AI companions can assist players in achieving their goals and enhancing the gaming experience, offering a valuable and practical solution for commercial FPS games. We developed BERT-FID, which incorporates the uncertainty of complex and fuzzy samples during training, to achieve effective collaborative inference with LLM in both accuracy and response performance. For scene recognition, we implemented a multi-modal dynamic entity retrieval scheme for 3D game assets, which effectively aligns human intentions with decision-making elements. A real-world user study demonstrated that ACVI can effectively understand 87.2% natural language commands and identify dynamic entities in the game. The improvements in instruction reasoning and scene recognition reduce inference deployment costs by approximately 77%. ACVI offers a comprehensive pipeline reference for designing AI teammates in commercial games, which is also applicable to a variety of virtual games, including RPGs (Role-Playing Games) like The Legend of Zelda and MOBA (Multiplayer Online Battle Arena) games such as Honor of Kings. Developers only need to prepare the BERT dataset, the LLM prompts, and the game asset dataset. This approach will facilitate more engaging and realistic interactions between players and AI characters. In the future, we will explore the relationship between audio signals and player intentions to enhance voice processing and reduce reasoning latency. Additionally, we aim to equip AI companions with distinct personality traits to influence their autonomous behavior, thereby further enhancing player enjoyment.

# 8    ETHICS STATEMENT

Informed consent was obtained from all participants prior to their involvement in the user study. Participants were made aware of the research's purpose, the procedures involved, and any potential risks and benefits. All data collected from participants were anonymized to ensure confidentiality, with access restricted solely to the research team.

We recognize the potential for our research to influence real-world applications, particularly in military contexts. ACVI relies on access to built-in APIs, such as behavior trees, in-game state information, and game resources. Therefore, it cannot be directly applied to real-world robotics settings or defense-related scenarios. In the gaming world, the behavior of NPCs is determined by the developers' intentions. The behavior and feedback of the AI have undergone rigorous review to ensure that the game content aligns with social and ethical standards.

We are committed to maximizing the benefits of our research while minimizing any potential harm. The AI companions are designed to provide a positive and engaging experience for players. We will conduct thorough testing and gather feedback through regular player surveys, community discussions, and game analyses to ensure that the AI behavior within the game does not promote or encourage real-world violence or aggressive behavior. Additionally, any potential psychological impacts on players will be carefully monitored and addressed.

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

# A    APPENDIX

## A.1    DETAILS OF ACVI

The pseudocode for handling fuzzy data during the training of BERT is illustrated in Algorithm 1 and the pseudocode for instructional reasoning and muti-modal dynamic search is illustrated in Algorithm 2.

---

**Algorithm 1** Entropy Optimization for Unsupported Intent Classification

---

Batch of samples $X = \{x_1, x_2, \ldots, x_N\}$, where $N = N_C + N_e$.
$N_c \leftarrow$ Number of supported samples
$N_e \leftarrow$ Number of unsupported samples
Initialize weights $w_j$ for each category based on intent label statistics
**for** *each category $j$* **do**
  └ Calculate weight $w_j$ as the ratio of supported samples to the number of samples in category $j$
**Calculate Cross-Entropy Loss for Supported Data:**
Initialize $\mathcal{L}_{ce} \leftarrow 0$
**for** *each supported sample $i$* **do**
  **for** *each category $j$* **do**
    └ Update $\mathcal{L}_{ce}$ using the weight $w_j$ and the predicted probability $p_{ij}$
**Calculate Entropy Loss for Unsupported Data:**
Initialize $\mathcal{L}_{ent} \leftarrow 0$
**for** *each unsupported sample $i$* **do**
  Calculate the entropy based on the predicted probability vector $p_i$.
  └ Update $\mathcal{L}_{ent}$ with the calculated entropy
**Combine Losses:**
Set the final loss $\mathcal{L}$ as the sum of cross-entropy loss and scaled entropy loss.
Training the hierarchical intent classification task for BERT-FID

---

---

**Algorithm 2** Instructional Reasoning and Multi-modal Dynamic Search

---

**Require:** Fast process model BERT-FID $F_{bert}$, generative language model $F_{llm}$, threshold $\delta_1$, text similarity calculator $F_{text}$, text to image similarity calculator $F_{image}$, threshold $\delta_2$.
**Input:** query from human player $x$ (string format).
**Output:** executable action $A$ encoding sequence for game engine.
Get intent and target by $F_{bert}$, and compute the overall confidence $C_{bert}$.
**if** $C_{bert} < \delta_1$ **then**
  └ Get intent and target by the generation of $F_{llm}$.
Set action $A$ to include receiver, decision, posture type, move type, etc. **if** *has target* **then**
    Retrieve the top-k data pairs from the asset dataset using $F_{text}$.
    Compute the confidence $C_{text}$.
    **if** $C_{text} < \delta_2$ **then**
      └ Retrieve the top-k data pairs from the asset dataset using $F_{image}$.
    Reorder the retrieved assets based on the dynamic game information includes position, orientation, bounding box, cover point, etc.
    Set the specific target location to action $A$.

---

These are specific values that indicate the percentage of requests that are completed within a certain time frame. For example, the 90th percentile response time means that 90% of the requests were completed in that time or less.

## A.2    HUMAN EVALUATION

To continuously assess the capabilities of ACVI and gather player data, we have established a human testing and evaluation system as Fig. 8 that leverages real-time feedback from players. Specifically

in Fig. 9, annotators can evaluate various modules of ACVI, including ASR, intent recognition, and environmental perception. We have introduced a tagging system for the entity recognition module, which further enhances the data quality of the game's 3D assets. Additionally, we can link the corresponding command information to the game's tactical replay, making it easier to verify whether AI behaviors are executed accurately.

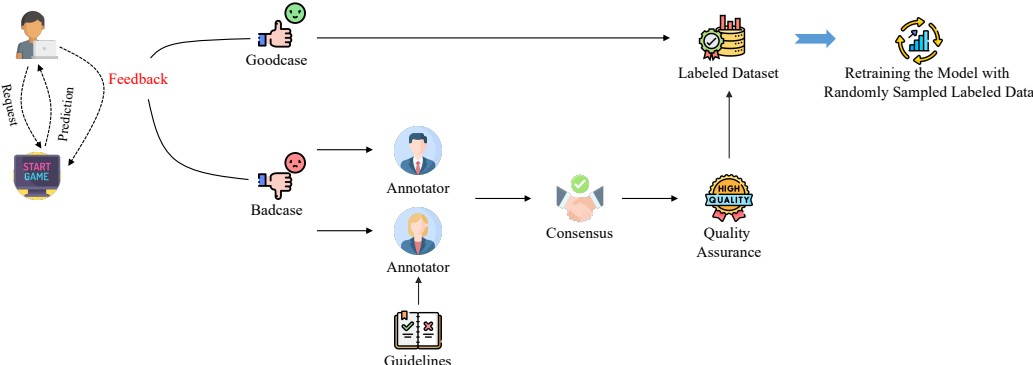

Figure 8: **The game testing system.** Human players can mark their real preference during gameplay. Good cases will automatically generate a dataset that can be used for model training and evaluation. For bad cases, we will collect dynamic game data and tactical replays used to the annotation system.

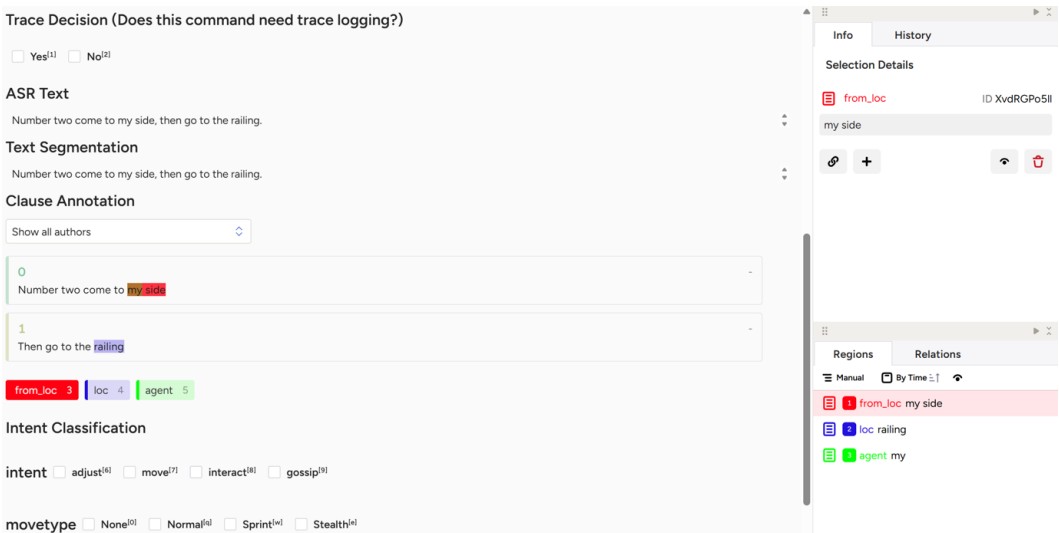

Figure 9: **The interface of the annotation system.** Annotators can evaluate various modules of ACVI, including ASR, intent recognition, and environmental perception.

## A.3 CASE STUDY

In order to conduct a better qualitative analysis of our system, we examine the capabilities of the ACVI system in handling hierarchical entity retrieval, sequential actions, task decomposition, and reference resolution, all within a complex interactive environment.

**Hierarchical Entity Retrieval.** In the first scenario, ACVI needs to retrieve a medkit from a couch. It uses BERT-FID to identify the target (move_to(target = ["couch", "medkit"])) and employs multimodal dynamic entity retrieval to find both the couch and the medkit in the environment. This approach allows the system to effectively navigate and interact with multiple dynamic objects in real time, improving its capability to perform complex retrieval tasks.

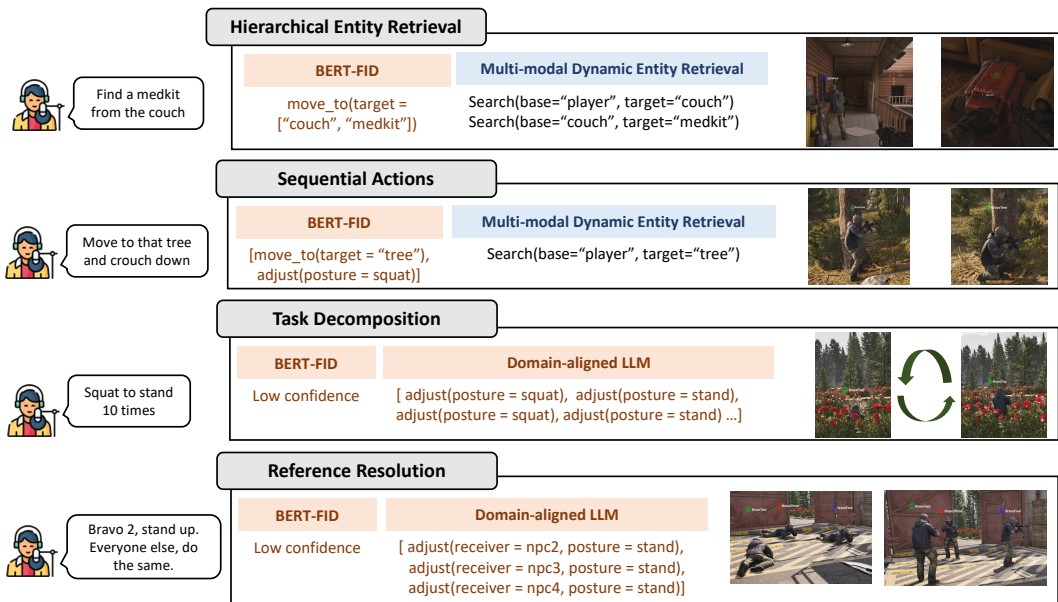

Figure 10: Four typical cases illustrate how the ACVI system accomplishes complex tasks, with the BERT-FID, LLM, and entity retrieval modules collaborating to provide real-time behavioral responses in the game.

**Sequential Instruction.** When instructed to "Move to that tree and crouch down," ACVI efficiently combines both the movement (move_to(target = "tree")) and posture adjustment (adjust(posture = squat)) into a single sequential action. The system's entity retrieval capabilities allow it to dynamically adjust to environmental changes, ensuring smooth execution of multi-step commands.

**Task Decomposition.** ACVI also excels at breaking down complex commands into manageable tasks. For instance, when asked to "Squat to stand 10 times," the system identifies the need for repeated posture changes and executes this action iteratively. In cases where the confidence level of BERT-FID is low, the domain-aligned LLM steps in to ensure precise task decomposition, guiding the agent through alternating postures to complete the task.

**Reference Resolution.** In the final scenario, ACVI processes an ambiguous command, "Bravo 2, stand up. Everyone else, do the same." Here, The domain-aligned LLM takes over to assign the correct actions to specific entities (adjust(receiver = npc2, posture = stand) for Bravo 2 and subsequently applying the same action to all other NPCs). This showcases ACVI's ability to handle reference resolution in multi-agent settings.

Overall, this case study demonstrates ACVI's proficiency in navigating hierarchical entity retrieval, performing sequential actions, decomposing complex tasks, and resolving ambiguous references, which are essential for enhancing the interactive depth of gaming environments.

Additional example interactions and their analysis is available in the Appendix A.4.

## A.4 SUCCESSFUL CASES

Table 6 presents various successful interactions between a player and an AI in a gaming context, categorized into four types: Simple Actions, Sequential Actions, Commands Reasoning, and Information Feedback.

In the "Simple Actions" section, the AI responds to straightforward commands from the player, such as following, moving to specific locations, or retrieving items like water. The "Sequential Actions" category showcases more complex directives where the player instructs the AI to perform tasks in a specific order, such as moving to a location and then returning. The "Commands Reasoning" section highlights the AI's ability to interpret the player's needs based on context, providing assistance like

Table 5: Additional successful cases in game.

| Interaction Type | Human-AI Interaction |
|---|---|
| Simple Actions | Player: Stay on me.
AI: Copy that.
AI: <follow(player) > |
| | Player: Check the blue box on the left.
AI: Roger that.
AI: <move_to("left, blue box")> |
| | Player: Give me a bottle of water.
AI: Copy.
AI: <move_to(player), item("water")> |
| | Player: smoke the exfile point.
AI: Roger that.
AI: <move_to("escape point"), item("smoke")> |
| Sequential Actions | Player: Move to the green car, and then come back.
AI: Copy that.
AI: <move_to(green car), follow(player)> |
| | Player: Place a medkit at the nearby cover location.
AI: Roger that.
AI: <move_to(coverpoint), item(medkit)> |
| | Player: Hold your fire, keep going!
AI: Copy that.
AI: <adjust(not fire), move_to(coverpoint)> |
| | Player: Cover fire in your position, I flank them.
AI: Engaging.
AI: <attack(enemy)> |
| Commands Reasoning | Player: I'm hurt.
AI: Here is a medkit for you.
AI: <move_to(player), item(medkit)> |
| | Player: I'm thirsty.
AI: Here is a bottle of water for you.
AI: <move_to(player), item(water) > |
| | Player: I'm low on health.
AI: A medkit has been placed for you.
AI: <item(medkit) > |
| | Player: I'm surrounded!
AI: <protect(player) > |
| | Player: Something's not right about this door.
AI: I'll check out.
AI: <move_to(door)> |
| | Player: Go to the motel.
AI-2: <move_to(motel)>
Player: Come Back.
AI-2: <move_to(player)>
Player: Go to the place I just asked you to go.
AI-2: <move_to("motel") > |
| Information Feedback | Player: What does the red smoke represent?
AI: The red smoke is an evacuation point. You can escape from here. |
| | Player: How to get out?
AI: Find the extraction point, and survive. That's the only way you're getting out of here. |
| | Player: Loot the body for me.
AI: Negative. It's all yours, I'll cover you. |
| | Player: Give me a key to the main guest room.
AI: Negative. That key costs 900K. I can't afford it. |
| | Player: Bravo2, attack Bravo 3
AI: Negative. No friendly fire! |

Table 6: Additional successful cases in game, Chinese version.

| Interaction Type | Human-AI Interaction |
|---|---|
| Simple Actions | Player: 保护我
AI: 收到。
AI: <follow("玩家") > |
| | Player: 去我左边的蓝色箱子看下。
AI: 收到。
AI: <move_to("左，蓝色箱子")> |
| | Player: 给我一瓶水。
AI: 收到。
AI: <move_to("玩家"), item("水")> |
| | Player: 给撤离点封烟！
AI: 收到。
AI: <move_to("撤离点"), item("烟")> |
| Sequential Actions | Player: 你走到绿车那里再回来。
AI: 收到。
AI: <move_to("绿车"), follow("玩家")> |
| | Player: 在附近找个掩体放置医疗包。
AI: 收到。
AI: <move_to("掩体点"), item("医疗包")> |
| | Player: 打枪的不要。
AI: 收到
AI: <adjust("闭火"), move_to("掩体点")> |
| | Player: 开枪掩护，我去绕后。
AI: 正在推进。
AI: <attack("敌")> |
| Commands Reasoning | Player: 我受伤了。
AI: 已放置医疗物资。
AI: <move_to("玩家"), item("医疗包")> |
| | Player: 我渴了。
AI: 已放置矿泉水。
AI: <move_to("玩家"), item("水") > |
| | Player: 我没血了。
AI: 已放置医疗包。
AI: <item("医疗包") > |
| | Player: 我被包围了。
AI: <protect("玩家") > |
| | Player: 门口有情况。
AI: 正在前往侦察。
AI: <move_to("门口")> |
| | Player: 前往汽车旅馆。
AI-2: <move_to("汽车旅馆")>
Player: 回来。
AI-2: <move_to("玩家")>
Player: 去刚才我让你去的地方。
AI-2: <move_to("汽车旅馆") > |
| Information Feedback | Player: 那红色烟雾是什么？
AI: 红色烟雾代表撤离点，你可以从这里带出你所需要的物资。 |
| | Player: 怎么从战场撤离？
AI: 寻找撤离点，这是你离开这里的唯一方法。 |
| | Player: 帮我舔包。
AI: 战利品属于你，我可以帮你架枪。 |
| | Player: 给我一把主客房钥匙。
AI: 我没有钥匙，你可以从局外大厅购买。
Player: 二号攻击三号！
AI: 不可攻击队友。 |

delivering medkits or protecting the player when in danger. Lastly, the "Information Feedback" category illustrates the AI's role in providing critical information and responses to player inquiries, such as explaining the significance of game elements or denying requests based on game mechanics.

Overall, the table emphasizes the dynamic and responsive nature of human-AI interactions in the game, showcasing the AI's capabilities in understanding and executing player commands effectively.

### A.5    FAILURE CASES

Despite its strengths, ACVI has its limitations. The following examples illustrate scenarios where the system may struggle to accurately interpret or execute commands, potentially leading to failures:

**Unresolvable Commands.** Due to the limitations of the underlying implementation logic, the response actions in the behavior tree are finite. For example, if a player asks the AI to help with a countdown, but the countdown function is not pre-defined in the behavior tree interface, the AI can only refuse the command.

**Spatial Awareness in Architectural Contexts.** Current text-image retrieval models struggle with the three-dimensional perception of space. For example, if a player''s target is "the second window from the left on the second floor," we cannot effectively retrieve this based on spatial relationships. We hope to establish a deeper connection between 3D models and text matching.

These failure cases highlight the critical opportunities and challenges for AI teammate systems. By identifying and analyzing these issues, we can focus efforts on enhancing ACVI's performance and increasing user satisfaction in future iterations.

