# OpenReview forum: "Revolutionizing AI Companion in FPS Games"
_ICLR.cc/2025/Conference — ICLR 2025 Conference Withdrawn Submission_

### Official Review · Reviewer_ZuPJ · 2024-10-26

**Soundness:** 2
**Presentation:** 2
**Contribution:** 2
**Rating:** 5
**Confidence:** 3

**Summary:**

This work proposes a system named "AI Companion with Voice Interaction" (ACVI), which is an AI system in the first-person shooter game "Arena Breakout: Infinite" that can be controlled by players' commands. The system consists of a fine-tuned BERT model that can decode simple commands, a domain-aligned LLM that is distilled from SOTA LLMs in order to process more complex commands, and a dynamic entity retrieval module that uses the game engine to find relevant objects and locations. The system has high throughput on modern GPUs while not sacrificing much accuracy, making it practical for real-time game companions.

**Strengths:**

- The work is clearly ambitious, creating AI agents that follow human commands in an FPS game. In particular, this idea of real-time AI companions that follow user instructions in a game seems new to me.
- The results section demonstrates the speed and accuracy tradeoffs of the proposed system against simple baselines.
- The figures and diagrams are very helpful for summarizing the presented techniques.

**Weaknesses:**

- A core component of the functionality of this system relies on deep integration within the game engine in order to retrieve entities and get 3D coordinate information. This means that the proposed technique may not transfer to different games or real-world robotics settings.

- The entropy optimization objective needs more justification. I would expect a citation to prior work that justifies this choice or significantly more experimental data demonstrating that the model truly reduces the confidence level beyond predicted categories. There are many more common approaches in the literature around calibrating model uncertainty in unseen settings (such as disagreement between independently trained models, random network distillation, and other approaches), and this work does not compare against any of these approaches. Furthermore, it is unclear if "low confidence" settings should instead be handled as another possible category, explicitly asking the domain-aligned LLM to generate instructions instead of relying on uncertainty.

- Given that a major contribution of this paper is to speed up inference by using small models and large models, there should be mentions of this line of work in the related work section. To name just a few recent works: Leviathan et. al. "Fast Inference from Transformers via Speculative Decoding" and Chen et. al. "Improving Large Models with Small models: Lower Costs and Better Performance."

- Despite having "voice" as part of the method's name, the actual role of voice is insignificant as it directly gets translated to text through a general ASR module.

- The language used in this paper is often not appropriate for academic writing. For instance, "this revolutionary feature aims to create the most immersive experience for its player" (lines 75-76) sounds more like a line from an advertisement than a research article. Furthermore, there are many grammatical errors and typos throughout the paper (to name a few: "propose" is repeated on line 79, "involves" should be "involving" in line 206).

**Questions:**

- Are the "images" used in the CLIP search static assets from the game itself or are they rendered from the view of the AI agents?
- Is the human instruction data in Chinese or English? I ask because the pre-trained models used for BERT and CLIP seem to be the Chinese variants, but all the examples are written in English.

---

### Official Review · Reviewer_iEnZ · 2024-10-28

**Soundness:** 2
**Presentation:** 2
**Contribution:** 2
**Rating:** 5
**Confidence:** 3

**Summary:**

This paper introduces ACVI, an AI system integrated into the FPS game Arena Breakout: Infinite, enabling human players to interact with AI teammates through real-time voice commands. ACVI is built from several key modules, including BERT-FID, domain-aligned LLM, and multimodal entity retrieval. Experiments show that ACVI can accurately understand complex voice commands and identify many objects with low latency.

**Strengths:**

- The intention detection module and entity retrieval component demonstrate high accuracy.
- By combining these components, ACVI functions as a real-time system.
- ACVI has been tested in a real-world FPS game environment.

**Weaknesses:**

1. The title’s use of “revolutionizing” may overstate the contributions. Using a smaller model to reduce latency and turning to LLMs when confidence is low is not a novel idea. The domain-aligned LLM is basically a finetuned LLM, and the entity retrieval is just combining heuristics with a finetuned CLIP model.
2. The experimental setup mainly focuses on ablation studies for each module, with limited discussion of the overall system evaluation, making it unclear what AI companion baselines are used for comparison. Human studies comparing ACVI to baseline systems would be valuable for human-AI collaboration, while the paper only discussed user studies of command understanding and entity retrieval accuracy with one sentence.
3. The results do not demonstrate ACVI’s applicability in other game environments. The authors should discuss what design choices are game-specific, and how to generalize those modules to other game environments.
4. The presentation can be improved. For example, there are typos (minor):
  - line 321 “fist” -> “first”
  - many missing spaces in citations such as “CLIP(Radford et al., 2021)” -> “CLIP (Radford et al., 2021)”

**Questions:**

How can ACVI work without access to internal information such as behavior trees and 3d assets?

---

### Official Review · Reviewer_d5jE · 2024-10-30

**Soundness:** 3
**Presentation:** 3
**Contribution:** 2
**Rating:** 5
**Confidence:** 4

**Summary:**

This work introduces AI Companion with Voice Interaction (ACVI), the first-ever AI system that allows players to interact with FPS AI companions through natural language in the commercial FPS game, Arena Breakout: Infinite. The AI teammates can follow and response to human player’s commands through real-time voice interactions. ACVI uses a fine-tuned BERT model to do command recognition. If the confidence of the BERT model is low, a fine-tuned will take the job and do the action planning. They used a fine-tuned CLIP model to do the entity retrieval. Empirical studies show that ACVI achieves a low-cost and low-latency performance compared to baseline methods.

**Strengths:**

1. The paper is straightforward and easy to follow.
2. The paper proposes a feasible real-time low-cost human-AI interaction solution, which has already been deployed to commercial FPS game. The solution may contribute to more video games and related domains.

**Weaknesses:**

1. The paper suffers from its novelty. It mainly applies the combination of existing mature techniques, which prevents the paper from being a great work (but still has the potential to be a decent work).
2. The proposed method can only be applied to a limited domain, where it highly relies on access to the built-in APIs, like the behavior tree, in-game state information and game resources. It means the method can only be applied to open-source games or
used by game developers.
3. As a human-AI interaction system, just quantitive results are not enough. Feedback form human players need to be reported, which requires a user study with questionnaires.
4. Figure 3 is very confusing, in items of the rows of Segment, Entities and Syntactic, which need a longer caption to introduce and explain.
5. Variance of QPS and latency need to be reported in Table 3.
6. Though I know most of the players of this game are Chinese. It is still quite strange that ACVI uses Chinese-CLIP/BERT/Embedding without any explanation and all the examples and the provided video are shown in English.

Minor issues
1. Line 79, typo: “we propose proposed.”
2. Line 321, typo: “we fist”
3. Table 3 is not referred by any text.

**Questions:**

1. What does the open-ended commands mean in line 78? The agent seems to only be able to execute the commands provided by the behavior tree.
2. As a commercial game, will the model be deployed on the player’s client-side or on the server cloud? It is unlikely for the player to have 2 3080 and Tesla T4 GPUs locally. However, these GPUs are far from enough to be deployed on the server side.
3. What the performance of Qwen2-1.5B without domain-aligned in Table 1?
4. As shown in Table 4, is all the text replied by AI generated by the LLM? LLMs seem to reply much more frequently than the results shown in Table 1.
5. Table 3 is confusing. Does it mean that the Qwen2-7B processes 46 requests per second?
6. The provided video was recorded in an experimental condition, where the player is experienced and his articulation is exceptionally clear. Could you provide more videos that can reflect the real situations with real game players?

---

### Official Review · Reviewer_da6F · 2024-11-03

**Soundness:** 2
**Presentation:** 1
**Contribution:** 2
**Rating:** 3
**Confidence:** 3

**Summary:**

This paper addresses the problem of enabling end-users to use their voice to communicate and coordinate activities with AI teammates in the context of a first-person shooter (FPS) game. The paper presents the AI Companion with Voice Interaction (ACVI) and deploy it in the FPS game, Arena Breakout: Infinite. The approach receives a voice command, converted to text, and calls BERT-FID (Bert designed for Fuzzy Intent Detection) to either execute the command if it is simple or pass the command to a domain-aligned LLM if not. This switching mechanism is designed to balance performance parameters (accuracy in executing commands vs. computation time). Benchmarking is performed, including ablation studies, and looks at various performance metrics. The approach also includes a dynamic entity retrieval component that helps to determine where specific objects are in the environment to ground the action commands.

**Strengths:**

- The topic is relevant to ICLR (prompt engineering for novel tasks and for embodied AI)

- The paper develops a system that reports a novel capability for human-AI teaming in a multi-agent setting

- The paper includes ablation studies and looks at multiple metrics to analyze the approach.

**Weaknesses:**

-The graphical illustrations are nice and some of the text is helpful in the paper, but there is a lack of specificity (e.g., pseudocode) for applicability.

-The results section (6.3) is not written well. It would be better to establish and answer key research questions. Right now, it is a short collection of tables. Metrics are not clearly enumerated and discussed (e.g., accuracy, parameters (though, this one is more obvious), and "processing sample count." Words used to describe the table, such as "route" and "processing sample count" don't appear elsewhere, making it hard to understand these terms. Overall, the simple paragraph of "The results in Table 1...improving accuracy" is inadequate for buttressing the weight of the primary claims of the paper -- that the approach works and works well.

-The Top-k is reported in Table 2 for k = 5,10, and 20. That's ok, but that is not good enough to assess performance for a user in a game -- you need k = 1. It would be helpful if the authors included the Top 1 performance.

-All results should show standard deviation where appropriate (e.g., Latency for Table 3).

-The results are missing an evaluation showing that users can play this game and outperform a team of humans or AIs, etc. As such, it is unclear whether this approach is good enough or is inadequate.
-There are numerous

-The ethics of helping people play an FPS with AI is a topic that should have been discussed in this paper. It may be a nice intellectual problem, but the point of this paper is to help develop a system that will kill people (virtual or real). Many papers are sponsored by defense entities, and there is no ban on such work. However, it would have been appropriate for the authors to discuss the ethical theory that supports this work.

-The contribution to multi-modal document retrieval is unclear. There are no clear baselines against competitive prior work that are not just simple versions of CLIP. Perhaps the authors could explore these:

Wei, C., Chen, Y., Chen, H., Hu, H., Zhang, G., Fu, J., Ritter, A. and Chen, W., 2023. Uniir: Training and benchmarking universal multimodal information retrievers. arXiv preprint arXiv:2311.17136.

Wu, H., Mao, J., Zhang, Y., Jiang, Y., Li, L., Sun, W. and Ma, W.Y., 2019. Unified visual-semantic embeddings: Bridging vision and language with structured meaning representations. In Proceedings of the IEEE/CVF Conference on Computer Vision and Pattern Recognition (pp. 6609-6618).

Li, G., Duan, N., Fang, Y., Gong, M. and Jiang, D., 2020, April. Unicoder-vl: A universal encoder for vision and language by cross-modal pre-training. In Proceedings of the AAAI conference on artificial intelligence (Vol. 34, No. 07, pp. 11336-11344).

-It is unclear how ACVI chooses which of the four mechanisms in 6.1 to choose.

-"propose proposed" -- double word

**Questions:**

-Can the authors provide a clear enumeration of metrics with definitions?

-Given that the Domain-aligned LLM is only 1.5B parameters and the BERT-FID with Domain-aligned LLM is 1.6B parameters, the computation time is likely comparable. If the performance of the Domain-aligned LLM is 2% better, then why not just use that LLM? The paper seems to argue that BERT-FID is faster, and Table 1 highlights that the processing sample count shows a "better" (unclear if it is better) in terms of more for BERT and the LLM (wanting more for BERT and fewer for the LLM); however, this is unclear. Can this be explained?

-Can the authors provide additional supplementary details to state more precisely how this system works as well as provide code?

**Details Of Ethics Concerns:**

The paper develops an AI system to help a user kill people in a videogame, but I did not see a discussion of the ethical theories supporting the work.

---

### Comment · Area_Chair_gQeP · 2024-11-24

Dear reviewers,

Thank you for your diligent work on the reviews. Currently the paper has split scores: 3 by da6F and iEnZ, 5 by d5jE and ZuPJ. The authors have responded to every single one of the reviews.

Reviewers da6F and iEnZ: did the authors' rebuttals and other reviews affect your score? Please respond before the 26 November to let the authors have time to respond if you need any clarification. Thank you!

Your AC

---

### Note · Authors · 2025-01-02

I have read and agree with the venue's withdrawal policy on behalf of myself and my co-authors.